Feature selection based on differentially correlated gene pairs reveals the mechanism of IFN-β therapy for multiple sclerosis

Jin Tao 1
Wang Chi 2
http://orcid.org/0000-0002-5942-1542 Tian Suyan 3 windytian@hotmail.com
1 Department of Neurology and Neuroscience Center, The First Hosptial of Jilin University , Changchun , China
2 Department of Biostatistics and Markey Cancer Center, University of Kentucky , Lexington, KY , USA
3 Division of Clinical Research, The First Hosptial of Jilin University , Changchuan, Jilin , China
Maggi Laura
Electronic publication date: 2020 Mar 16
Publication date: 2020
Volume: 8
Electronic Location ID: e8812
Received 2019 Sep 25; Accepted 2020 Feb 27
Copyright: © 2020 Jin et al.
Copyright year: 2020
Copyright holder: Jin et al.
License: This is an open access article distributed under the terms of the Creative Commons Attribution License, which permits unrestricted use, distribution, reproduction and adaptation in any medium and for any purpose provided that it is properly attributed. For attribution, the original author(s), title, publication source (PeerJ) and either DOI or URL of the article must be cited.
License URL: https://creativecommons.org/licenses/by/4.0/

Keywords: Multiple sclerosis, Differentially correlated edge, Longitudinal, IFN-beta therapy, Treatment response, Feature selection

Funding: Education Department of Jilin Province JJKH20190032KJ Natural Science Foundation of China 81671177 and 31401123 Natural Science Foundation of Jilin Province Science and Technology Development Plan Project 20190201043JC This study was supported by the Education Department of Jilin Province (grant No. JJKH20190032KJ), the Natural Science Foundation of China (grants Nos. 81671177 and 31401123), and the Natural Science Foundation of Jilin Province Science and Technology Development Plan Project (No. 20190201043JC). The funders had no role in study design, data collection and analysis, decision to publish, or preparation of the manuscript.

==============================
Multiple sclerosis (MS) is one of the most common neurological disabilities of the central nervous system. Immune-modulatory therapy with Interferon-β (IFN-β) is a commonly used first-line treatment to prevent MS patients from relapses. Nevertheless, a large proportion of MS patients on IFN-β therapy experience their first relapse within 2 years of treatment initiation. Feature selection, a machine learning strategy, is routinely used in the fields of bioinformatics and computational biology to determine which subset of genes is most relevant to an outcome of interest. The majority of feature selection methods focus on alterations in gene expression levels. In this study, we sought to determine which genes are most relevant to relapse of MS patients on IFN-β therapy. Rather than the usual focus on alterations in gene expression levels, we devised a feature selection method based on alterations in gene-to-gene interactions. In this study, we applied the proposed method to a longitudinal microarray dataset and evaluated the IFN-β effect on MS patients to identify gene pairs with differentially correlated edges that are consistent over time in the responder group compared to the non-responder group. The resulting gene list had a good predictive ability on an independent validation set and explicit biological implications related to MS. To conclude, it is anticipated that the proposed method will gain widespread interest and application in personalized treatment research to facilitate prediction of which patients may respond to a specific regimen.

Introduction

Multiple sclerosis (MS) is an immune-mediated, inflammatory demyelinating disease of the central nervous system that affects about 2.3 million people worldwide (Browne et al., 2014; Brownlee et al., 2017). The majority of patients are women between 20 and 40 years old, and the ratio of female to male is about 3:1 (Browne et al., 2014; Brownlee et al., 2017). Depending on the initial course of disease, MS may be categorized as relapsing-remitting or primary progressive disease (Browne et al., 2014). Relapsing-remitting MS (RRMS) is the most prevalent form of MS, accounting for 85% of MS cases. As indicated by its name, it involves relapses followed by series of remission. In the early stage, patients with this type of MS may experience complete recovery or partial sequel; however, approximately 50% of patients progress over time to secondary progressive disease, with or without acute relapses. In contrast, primary progressive MS begins with gradual neurologic deterioration from onset (Browne et al., 2014; Vargas & Tyor, 2017).

The etiology of MS has not been fully elucidated, however, it is commonly believed to be triggered as an autoimmune response to an interaction between genetic and environmental factors (Nafee, Watanabe & Fregni, 2018). In the pathogenesis of MS, T helper (Th)1 cells release pro-inflammatory cytokines such as interleukin-2 (IL-2), interferon-gamma (IFN-γ) and tumor necrosis factor-α (TNF-α) and play a crucial role in inflammation of MS. Th17 cells release IL-17 and have some impacts on MS as well. On the other hand, Th2 cells secrete IL-4, -5 and -10 to counter-regulate Th1 and Th17 responses (Garg & Smith, 2015). Interferon-β (IFN-β) is the first class of treatment. It was approved for RRMS in 1993, and was subsequently approved in Europe for secondary progressive MS with continued relapse. It is thought that IFN-β works to combat MS by stabilizing the blood brain barrier and altering relevant cytokines by redirecting pro-inflammatory Th1 and Th17 immune responses into anti-inflammatory Th2 responses.

There are an increasing amount of studies suggesting that genomic data such as gene expression profiles and genetic variants data may provide insightful clues on the development and molecular subtypes of MS. For example, Parnell et al. (2014) tested the expression values of 110 known MS-related genes, and found that expression of the transcription factors (TF) controlling T and NK cell differentiation, EOMES, TBX21 and other TFs was significantly lower in MS compared to healthy controls. Besides the long-time well-known strong genetic effect of the HLA-DR2 locus, non-HLA variants (such as IL2R, IL7R and IL2R variants) have been implicated. Among them, IL2R and IL7R are focused the most since they are expressed on regulatory T cells which play crucial roles in regulating the immune response in MS patients (Nafee, Watanabe & Fregni, 2018).

Feature selection is a machine learning strategy of selecting, from among thousands of genes, a gene signature (subset) that may be relevant for diagnosis of a disease, segmentation of disease subtypes, patient drug response or survival rate prediction, and is becoming routine practice in the fields of bioinformatics and computational biology (Hira & Gillies, 2015; Ang et al., 2016). Until now, the majority of available feature selection methods have focused on investigating alterations of gene expression levels. These are referred to as node-centric methods. An alternative strategy is to focus on alterations of relationships (correlation) among genes, which should also bear meaningful information about which genes are likely to be associated with the outcomes of interest. Here, a “node” corresponds to a gene in a gene-to-gene interaction network/graph and an “edge” corresponds to the correlation between a gene pair. As pointed out by Kostka & Spang (2004), a differential expression analysis focuses on the first moments (i.e., means) and a differential correlation analysis aims at the second moments instead from the prospect of statistics. So far, development of such edge-centric methods is far from being comparable to development of node-centric counterpart (see (Kostka & Spang, 2004; Carter et al., 2004; Fuller et al., 2007) for examples of edge-centric methods).

Compared with classic feature selection methods which mainly deal with cross-sectional data, the feature selection process for longitudinal gene expression data is more complicated. This is unsurprising given the fact that longitudinal gene expression data involve more than a single time point and consider both the expression value trajectory over time and their differences between different phenotypes. So far, feature selection methods specifically to handle longitudinal data are far from sufficient, to the best of our knowledge, the method proposed by Sun et al. (2013) is one of the few edge-centric methods capable of dealing with longitudinal data. Specifically, in this so called differentially expressed network method, an edge is selected if the directly linked gene pair is strongly correlated (the absolute Spearman correlation coefficient is larger than a threshold) in one group while not in the other group. Combining the differentially expressed genes of the two groups builds a differentially expressed network. However, using a threshold to decide the connection status between gene pairs may result in many falsely positive edges in which the difference of correlation coefficients under two different conditions may be marginal.

Motivated by the differentially expressed network method (Sun et al., 2013), we proposed a bioinformatics procedure to identify gene pairs with differentially correlated edges (DCEs) over time by replacing the difference in connection statuses with that in actual correlation coefficients. Then we applied the proposed method to a longitudinal microarray dataset which evaluates the effect of IFN-β on MS patients. Our objective is to further elucidate the underlying therapeutic mechanisms of IFN-β and to predict which subset of MS patients will respond to IFN-β therapy based on the identified molecular signature.

Materials and Methods

Experimental data

A microarray experiment dataset was used as a training set in this study (GEO repository accession number GSE24427) (Hundeshagen et al., 2012). We collected longitudinal gene expression profiles from 25 German RRMS patients treated with recombinant IFN-β-1b (250 µg every other day) for 2 years. The chips were hybridized on the Affymetrix hgu133A and B platforms. Only chips hybridized on the Affymetrix hgu133A platform were considered. This platform included 22,283 probe sets, corresponding to 12,437 unique genes.

Gene expression values were measured at five separate time points: before the first, second, 1-month, 12-month and 24-month injections. Keeping in mind that several patients may have relapsed shortly after the 2-year treatment (in which case their gene expression profiles may have undergone some changes similar to changes of non-responders), we restricted the responder category to patients whose first relapse time was more than 5 years (60 months), resulting in nine responders and nine non-responders fed into the downstream analysis.

To validate the predictive performance of the final model, data from the GSE19285 experiment (Hecker et al., 2012) were downloaded from the GEO repository to serve as a validation set. In that study, 24 RRMS patients were injected with intramuscular IFN-β-1a once a week and the gene expression data were collected at the first, second and fifth injections. Likewise, the chips of that study were hybridized on both the Affymetrix hgu133A and B platform, but only data on the Affymetrix hgu133A platform were considered in this study. By adding the restriction of more than 5-years duration for the first relapse, 21 patients were included in this testing set; 12 were responders and nine were non-responders.

Pre-processing procedures

Raw data (CEL files) of these two experiments were downloaded from the GEO repository, respectively. The expression values were obtained using the fRMA algorithm (McCall, Bolstad & Irizarry, 2010). The expression summary values were normalized using quartile normalization and then log2 transformed.

Statistical methods

Identification of consistent DCEs

The procedure to identify consistently DCEs comprised three major steps: (1) interactions in String software (Franceschini et al., 2013) were used as a reference network (which slightly downsized the number of genes under consideration from 12,437 to 11,502), then Spearman correlation coefficients (SCCs) were calculated separately for the responder group and the non-responder group at each specific time point; (2) when the absolute difference of SCCs between two groups at a specific time point was larger than a pre-determined threshold (i.e., 0.6, this value may corresponds to two possible situations: either the correlation directions in two groups are the same with in one group being extremely large whereas in the other group being small; or the correlations in two groups are at least moderate but in different directions. In addition, with this step the number of gene pairs under consideration for p-value calculation was downsized to a manageable scale), the corresponding gene pair was deemed to be a DCE at that time point; and (3) the intersection of DCEs across 5 time points was taken and the resulting subset was considered to represent consistently DCEs over time. A flowchart of the procedure is elucidated using a toy example with five genes (Fig. 1).

Figure 1 Flowchart of the proposed procedure.

(A) Identification of differentially correlated edges. (B) Generation of consistently differentially correlated edges. Here, a toy-example with 5 genes was used to graphically elucidate the proposed procedure. First, a reference network was created according to the String software. Only the interactions recorded in this software were considered. Then the Spearman correlation coefficients for these gene pairs were calculated for the responder group and the non-responder group, respectively. If the absolute value of the correlation coefficient difference between these two groups was larger than a predetermined cutoff, say, 0.6, the corresponding edge was deemed as differentially correlated edge (DCE). Those common DCEs across time points were consistently DCEs. Then the corresponding p-values for these DCEs were calculated using permutation tests. Upon the genes involved in the statistically significant DCEs, the biological relevance was investigated.

Results

DCEs

In this study, the connection information in the String software was used as a reference. Only when the confidence score of a gene pair is larger than 0.6, these two genes are considered to be connected by an edge. The cutoff value for the absolute difference of SCCs between two groups is set at 0.6. Using the proposed procedure, 384 consistently DCEs over five time points were identified. These edges included 510 unique genes. By saying consistently, it is referred to the scenario that these edges were identified to be differentially correlated across all time points (i.e., the overlapped edges among these time points).

Using permutation tests and following the strategy proposed by Anglani et al. (2014) the significance levels of these gene pairs were determined and presented in Table 1 as well. Twenty-two DCEs are deemed to be statistically significant (with a p-value < 0.01), which involve 41 unique genes (Table 1). Using these 41 genes as predictors, a linear support vector machine model was fit to the training set. The resulting model was applied to the validation set (an independent dataset from the training set) to calculate posterior probability of a MS patient being a responder at each time point. With four responders and eight non-responders misclassified, a predictive accuracy of 80.95 % (51/63) was achieved by this 41-gene signature. Given that demographic heterogeneities may exist between patients in both experiments, and that the collection time points in the two studies differed hugely, with the early period of treatment being concentrated in the second study, this predictive accuracy is fairly good.

Table 1 Consistent edge differences (network differences) across five time points.

Gene1	Gene2	T1	T2	T3	T4	T5	p-Value	
R	NR	R	NR	R	NR	R	NR	R	NR	
ADCY1	AKT1	−0.567	0.65	−0.717	0.7	−0.383	0.667	−0.883	0.283	−0.683	0.167	0.0001	
FKBP10	HTR5A	0.633	−0.4	0.717	−0.433	0.833	−0.033	0.617	−0.217	0.967	−0.083	0.0087	
DCAF13	KRR1	−0.533	0.467	−0.133	0.6	−0.6	0.35	0.533	−0.5	−0.7	0.217	0	
ATRX	XRCC6	0.233	−0.533	0.283	−0.55	−0.75	−0.117	0.517	−0.467	0.567	−0.5	0.0095	
CSF2	CXCR3	−0.1	−0.883	0.3	−0.7	−0.033	−0.783	0.283	−0.6	0.333	−0.767	0.0083	
LIPE	PNLIP	0.033	0.783	0.533	−0.267	0.317	−0.567	0.8	0.1	0.433	−0.5	0.0059	
EP300	GTF2H5	0.933	−0.3	0.633	−0.267	0.817	−0.733	0.617	−0.5	0.817	−0.017	0.0024	
EP300	POLR2K	0.817	−0.35	0.783	−0.117	0.8	−0.733	0.65	−0.533	0.833	−0.167	0.0094	
UBE2D2	UBXN7	0.467	−0.367	0.467	−0.267	0.633	−0.533	0.617	−0.033	0.717	−0.15	0.0068	
ALMS1	CEP290	−0.683	0.25	−0.617	0.567	−0.717	−0.05	0.733	−0.617	0.783	−0.05	0.0047	
RSRC1	SNU13	−0.683	0.433	−0.683	0.433	−0.45	0.333	−0.633	0.667	−0.217	0.717	0.0023	
BCR	CUX1	0.217	−0.5	0.633	−0.767	0.533	−0.533	0.35	−0.65	0.333	−0.367	0.007	
FGF21	FLT1	0.6	−0.433	0.717	−0.517	0.183	−0.667	0.983	−0.417	0.833	−0.117	0.0014	
DYNC1H1	TUBGCP2	−0.7	0.3	−0.767	0.817	−0.767	0.317	−0.55	0.3	−0.467	0.517	0.0006	
CXCL9	GRM4	0.1	−0.85	0.05	−0.583	0.25	−0.5	0.767	−0.083	0.8	−0.167	0.0007	
DYNC1H1	PCNT	0.483	−0.75	0.4	−0.933	0.267	−0.65	0.567	−0.6	0.4	−0.367	0.004	
CROCC	PCNT	−0.467	0.617	−0.283	0.817	−0.45	0.683	−0.517	0.733	−0.483	0.233	0.0079	
IL2RA	IL2RB	−0.5	0.317	−0.583	0.017	−0.75	0.15	−0.783	0.517	−0.467	0.433	0.0072	
MELK	RFC4	−0.367	0.683	−0.117	0.533	−0.633	0.617	−0.033	0.583	−0.867	0.133	0	
HNRNPA2B1	RBM17	−0.733	0.233	−0.767	0.4	−0.617	0.2	−0.683	0.017	−0.9	−0.183	0.0026	
ADCY3	NUDT2	0.6	−0.267	0.483	−0.65	0.417	−0.667	0.65	−0.433	0.25	−0.517	0.0095	
AGA	GCA	0.817	−0.317	0.817	−0.133	0.8	−0.65	0.617	−0.617	0.633	−0.45	0.0099	
Note:

R, responders; NR, non-responders. The gene pairs fit two conditions: (1) The difference for Spearman correlation coefficient between responders and non-responders >0.6 and (2) the corresponding p-value < 0.01.

Discussion

The GeneCards database (Safran et al., 2010) indicates that of the 41-gene signature, 23 genes are directly associated to MS. Among these 23 genes, seven genes (i.e., CXCL9, IL2RA, CXCR3, AKT1, CSF2, IL2RB and GCA) have a confidence score >2 (Fig. 2). Likewise, a SVM model was fit with these seven genes as predictors and then applied to the validation set. This results showed that this seven-gene signature can achieve a predictive accuracy of 65.08 % (41/63) with six responders and 16 non-responders misclassified. Furthermore, these seven genes whose mean expression values for the responders and non-responders across time points are diagramed in Fig. 3. According to the EDGE method (Storey et al., 2005), only AKT1 (adjusted p = 0.003) and CXCL9 (adjusted p = 0.012) are significantly differentially expressed over time between these two groups. Focusing specifically on the two extreme points, no genes are differentially expressed at the first time point while only CXCL9 is under-expressed in the responder group at the last time point according to Wilcoxon’s tests.

Figure 2 Venn-diagram showing 41 genes involved in the statistically significant DCEs.

Among them, 23 genes are indicated to be directly related to multiple sclerosis, whose symbols are given in the Venn-diagram. 7 of 23 genes have a confidence score >2, which are highlighted in red. DCEs: consistently differentially correlated edges.

Figure 3 Mean expression value trajectories of the identified seven-gene list for the responder and the non-responder.

(A) For CXCL9. (B) For IL2RA. (C) For CXCR3. (D) For AKT1. (E) For CSF2. (F) For IL2RB. (G) For GCA. Here, R: the responder group; NR: the non-responder group.

Next, a gene-to-gene interaction network of these seven genes was constructed using the String software and presented in Fig. 4. In this figure, we found that these genes were highly connected with all genes but GCA (which is an isolated gene) circling around one another to frame a loop. Interestingly, it was observed that the edges between IL2RA and IL2RB, and the edge between CXCR3 and CSF2 were negatively correlated to each other in the responder group while positively correlated in the non-responder group. Further study is highly desirable to explore if such correlation patterns have any impact on how a MS patient responds to the IFN-β treatment.

Figure 4 Gene-to-gene interaction network of the identified gene list.

(A) The responder group. (B) The non-responder group. Here, + stands for the correlation coefficient between the corresponding gene pair is positive; − stands for the correlation coefficient between the corresponding gene pair is negative.

How these seven genes are related to MS was mined by a PubMed literature searching. Since the autoimmune inflammatory process is believed to be essential for the development of MS, it is natural to observe that chemokines and chemokine receptors are involved in the pathogenesis of the disease (Szczuciński & Losy, 2007). For instance, a study (Murzenok, Matusevicius & Freedman, 2002) compared the chemokine receptor—CXCR3—expression by gamma delta T cells derived from the blood and CSF of MS patients with health controls and observed its expression increased. Another study (Mahad et al., 2003) also suggested significantly different expression of CXCR3 on peripheral blood lymphocytes in MS compared with controls.

Furthermore, CXCL9 is a ligand of CXCR3. A study (Korniejewska et al., 2011) described two alternatively spliced variants of the human CXCR3-A receptor, termed CXCR3-B and CXCR3-alt. While human CXCR3-B binds CXCL9, CXCL10, CXCL11 as well as an additional ligand CXCL4, CXCR3-alt only binds CXCL11. A study by Mellergård et al. (2010) suggested there was a marked decline in CSF levels of cytokines and chemokines in MS patients treated with natalizumab, thus concluding chemokines associated with both Th1 (CXCL9, CXCL10, CXCL11) and Th2 (CCL22). However, the chemokine system is complex and consists of numerous ligands and receptors. It is highly unlikely that targeting a particular simple chemokine or chemokine receptor will give beneficial effects.

A recent review (Leray et al., 2016) stated that in addition to IL7RA, IL2RA is a well-known immunogenetic marker of MS, explaining why IL2RA was ranked on the top of the directly related to MS list given by the GeneCards database (Safran et al., 2010). Although the association between IL2RA and MS has been well established, the functional variation is still unknown. It is speculated that the effect of IL2RA on MS should be better described by several SNPs than by a single one (Babron et al., 2012), and they showed that the set of SNPs rs2256774 and rs3118470 provided a perfect discrimination between the MS patients and controls. Furthermore, there is a potential link between IL2RA and CXCR5 regarding MS. A study in China (Xia, Qin & Zhao, 2018) showed that the genotypes and allele frequency distributions at the loci of IL2RA rs2104286 and rs12722489 were significantly different between the MS and control groups. Meanwhile, the gene polymorphisms at the loci of IL2RA rs2104286 and rs12722489 may increase the onset risk of MS. IL2RA–rs2104286 showed a positive correlation with CXCR5–rs3922.

Using a meta-analysis, the association of polymorphisms in IL2RB, IL2RA and IL2 with MS were systemically reviewed, in which significant association for IL2RA and IL2 had been justified (Cavanillas et al., 2010) and the association for IL2RB was deemed to be insignificant. No significant changes in serum IL-2 and soluble IL-2R levels were found between MS patients and controls, either (Ott et al., 1993).

The expression of CSF2, also known as GMCSF, by human TH cells has been reported to be associated with MS disease severity. GMCSF is strongly induced by interleukin 2 (IL2) (Hartmann et al., 2014): an MS-associated polymorphism in IL2RA specifically increased the frequency of GMCSF producing TH cells. The IL2RA polymorphism regulates IL2 responsiveness of naive TH cells and their propensity to develop into GMCSF producing memory TH cells. AKT is a critical mediator of growth factor-induced neuronal survival in the developing nervous system. Survival factors can suppress apoptosis in a transcription-independent manner by activating the serine/threonine kinase AKT1, which then phosphorylates and inactivates components of the apoptotic machinery. AKT activation relies on the PI3K pathway, and AKT1 is recognized as a critical node in the pathway. Furthermore, a recent study (Ouyang et al., 2019) found that compared with wild-type mice, AKT1−/− mice exhibited improved experimental autoimmune encephalomyelitis (EAE), a mouse model of MS. At the cellular level, AKT1 seems to inhibit the proliferation of thymogenic regulatory T cells, thereby promoting Ag-specific Th1/Th17 response. Additionally, Barca’s study (Barca et al., 2003) found that IFN-β promotes astrocyte survival by stimulating the PI3K/AKT pathway, and thus concluded that the beneficial effect of IFN-β in MS may depend partially on its ability to protect astrocytes from apoptotic cell death at the early stage of MS.

As far as GCA is concerned, Martínez et al. (2008) examined four SNPs along the IFIH1-GCA-KCNH locus, rs13422767, rs2111485, rs1990760 and rs2068330 and reported an association of this locus with MS for the first time. Nevertheless, such an association had not been validated in another study (Couturier et al., 2009). The authors explained this discrepancy with three possible reasons: different ethnic background between the study population; different study designs used and the sample size issue.

Furthermore, all these seven genes except AKT1 and IL2RB are IFN-responsive genes according to the Interferome database (http://www.interferome.org/). In summary, further investigation is warranted to examine how IFN-β treatment establishes its effects by targeting these seven genes.

Conclusions

Our procedure identified consistent DCEs over time for a responder group in comparison with a non-responder group, and the resulting gene signature has explicit biological relevance to MS. Based on simple calculations of SCCs and their corresponding difference between two different phenotypes, our method is easy to comprehend and can be implemented by an entry-level statistician or a clinician. Therefore, we anticipate its widespread application in relevant research areas, to help researchers identify the underlying therapeutic mechanism of a specific regimen (as shown in the Supplemental Materials, there the proposed procedure was applied to analyze the gene expression data of GSE41846 (Nickles et al., 2013) in which treated MS patients and non-treated MS patients were compared) and predict which patients are more likely to respond.

The statistical procedure used in this study has several limitations. First, it excludes genes that are not in overlapping DCEs, therefore, valuable information may be overlooked, especially for the first time point. In the future, a way to use the first time point as the reference may be considered such that the overlap of other time points with this point will become the focus. Second, the training dataset had only 25 samples in total, and to eliminate possible ambiguous results caused by patients having relapses in the first 2–5 years, seven MS patients were excluded from the downstream analysis. This results in a smaller sample size. A large gene expression experiment with a better study design is highly desirable.

Supplemental Information

Supplemental Information 1 R codes for identifying differentially correlated edges for multiple sclerosis microarray data.

Click here for additional data file.

Supplemental Information 2 Further validation using another microarray dataset.

Click here for additional data file.

We thank Donna Gilbreath for scientific editing and assistance in manuscript preparation.

Additional Information and Declarations

Competing Interests

Author Contributions

Data Availability

The authors declare that they have no competing interests.

Tao Jin analyzed the data, authored or reviewed drafts of the paper, and approved the final draft.

Chi Wang analyzed the data, prepared figures and/or tables, authored or reviewed drafts of the paper, and approved the final draft.

Suyan Tian conceived and designed the experiments, analyzed the data, prepared figures and/or tables, authored or reviewed drafts of the paper, and approved the final draft.

The following information was supplied regarding data availability:

Microarray data are available at GEO: GSE24427 and GSE19285.

The R codes for the proposed procedure are available in the Supplemental Files.

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
