# Peer review of "Feature selection based on differentially correlated gene pairs reveals the mechanism of IFN-β therapy for multiple sclerosis"

_PeerJ, doi:10.7717/peerj.8812_

## Round 0.1 · original submission · Major Revisions

Considering that MS is a multifactorial disorder with a strong implication of genetic and immunological involvement, an association could be inevitable seen. In the paper there is no correction for multiple testing performed or if done not presented. Please, present these data in the revised version.

·

Basic reporting

The manuscript is interesting, well written and relevant. But a few points should be addressed.

Under the introduction it´s stated that MS is a immune-mediated disorder. There are known genotypes for subsets of MS this ought to be mentioned.

Under the section of Biological implications Genecards are extensible used. If considering the involvment of AKT1 in MS the association i based on one GWAS study with a single SNP showing association to AKT1 but additionally nummerous other genes f.x. IGHA1. What is the argument for especially focusing on AKT1?
In the litterature AKT1 is mainly associated to different cancer phenotypes; Cowden, Proteus, Gastric cancer and others please consider these aspects.
Concerning Genecards and MPO and C3, respectively, here I don´t see these genes mentioned. whereas MPO and C3 are mentioned as associated to MS in pubmed searches. But then again a multitude of genes have been correlated to MS. Both MPO and C3 has been associated to a multitude of immunological related phenotypes - is the association seen here a MS association or an association to inflammation/immunological disease?
The study is using correlation coefficients hence both p-values and values corrected for multiple testing should be presented and addressed in the discussion.

Experimental design

The experimental design is performed fully sufficient.

Validity of the findings

As mentioned under basic reporting I miss further arguments and considerations under the biological implications and discussion on the statistically significans of the findings..

Reviewer 2 ·

Basic reporting

no comment

Experimental design

The authors proposed a bioinformatics method to identify gene pairs differentially correlated over time by replacing the difference in mean gene expression with that in correlation coefficients. The method was applied to a longitudinal microarray dataset which evaluates the effect of INF-β on multiple sclerosis (MS) patients. The aim is to study the underlying therapeutic mechanisms of IFN-β and to predict which MS patients will respond to IFN-β therapy based on the identified molecular signature. The idea is well described but the too small sample size reduces the power of the study and increases the margin of error. The authors have to investigate if in GEO repository are present more gene expression data sets to be analyzed with their procedure concerning MS patients treated with Interferon. See Nickles D., Chen H., Li M., Khankhanian P., Madireddy L., Caillier S., Santaniello A., Cree B., Pelletier D., Hauser S., et al. Blood RNA profiling in a large cohort of multiple sclerosis patients and healthy controls. Hum. Mol. Genet. 2013;22:4194–4205.

Validity of the findings

I commend the authors because the manuscript is clearly written in professional and unambiguous english language but there are major problems concerning the statistical analysis of data which should be improved upon before Acceptance.
1. It is not clear how the authors choose to adopt a threshold of 0.8 for differential correlation. The selection of differentially correlated genes is the result of the application of a statistical hypothesis test. It is necessary to estimate a null distribution for the difference in correlation between 2 genes (for example by permutation of labels) and by exploiting the null distribution, it is possible to compute the p-value. If the p-value indicates strong evidence against the null hypothesis, you reject the null hypothesis. It is necessary to introduce a threshold on the p-value. Typically, a p-value ≤ 0.05 indicates a strong evidence to reject the null hypothesis (see Anglani R, Creanza TM, Liuzzi VC, et al. Loss of connectivity in cancer co-expression networks. PLoS One 2014;9:e87075).
2. Why the authors choose to use Pearson’s correlation coefficients instead of Spearman correlation that it is considered more appropriate for these analysis. Moreover, we can read in conclusions: “so one may argue that Spearman’s correlation coefficients between gene pairs may be more suitable”.
3. What is the meaning of p=0.2 for differential expression? Is p the p-value? if it is so, this theshold is not appropriate. The authors have to clarify the meaning of p and eventually repeat the analysis with a more suitable threshold.
4. The predictive accuracy at line 197 is not clear: is 63 the size of the set of both responders and non responders? At lines 132-133, the authors declare that 21 patients are in the testing set.
5. The results are not clear mostly in their interpretation at lines 249-255. The authors write:” if the correlation directions are positive for the first three gene pairs and negative for the last time point”. Do they mean that the correlations directions are positive in the first time points? More important, at line 255 they write “for a patient without such correlation”, but how is possible to measure a correlation for a single patient? For a specific time point we have a value for each gene, we can not evaluate a value of correlation. I can not understand these considerations that represent the meaning and the value of their work.
6. Are the genes found by the authors known IFN-responsive genes as recorded in the Interferome database (http://www.interferome.org/?

---

## Round 0.2 · Minor Revisions

Please, improve the discussion section in order to address Reviewer 2 comments. and to better fit with the results section.

·

Basic reporting

No comment

Experimental design

No comment

Validity of the findings

No comment

Additional comments

No comment

Reviewer 2 ·

Basic reporting

The paper is well written but the authors have to correct these little errors.

• It is not correct the legend of Figure 2 just above the picture: “Venn-diagram showing the overlapped genes between differentially expressed genes over time and genes involved in the consistently differentially correlated edges.” This is to be substituted by “Venn-diagram showing 41 genes involved in the statistically significant DCEs. “ that is the correct legend that you can read in the list at 448 row.

• In the table 1, the negative sign cannot be on the top of the number, try to put the sign before the number on the same line.

• At the 225-226 rows, it is not correct to write that negative correlation means that “one is overexpressed/under-expressed, then the other tends to be overexpressed/under-expressed as well.” This the case of positive correlations.

• In the legend of figure 3, it is to be correct 6-gene list with 7-gene list.

Experimental design

no comment

Validity of the findings

I noted that the discussion is not properly supported by the results section: some conclusions are not limited to supporting results.
• In details, the rows from 203 to 220 describe the differences in expression mean values between the two groups such as the following text in capitol letters: “From Figure 3, it is observed that at the first time point (the measure before IFN-β treatment was given) the expression levels of CXCL9, CSF2, IL2RB and GCA were BASICALLY THE SAME between the two groups; IL2RA and CXCR3 had a LOWER EXPRESSION value in the non-responder group; and AKT1 had a lower expression value in the responder group. At the last time point (the measures at year two when all non-responders had experienced the first relapse and the responders were at least three years away from their first relapses), CXCL9 and CSF2 had a lower mean expression value in the responder group; AKT1, IL2RA and CXCR3 had a HIGHER expression value in the responder group; and the mean expression values of IL2RB and GCA remained approximately identical.” These conclusions refer to the differential expression of genes between the two conditions and have a scientific meaning only if they follow a statistical analysis of differential expression that turns out a statistical significance of results but I do not understand why the authors chose to eliminate the analysis of differential expression in this second version of the paper.

• In the result section, at 186-189 rows we found the value of the performance of a support vector machine model trained by using the 41 genes as predictors to compute the probability of a MS patient being a responder at each time point (accuracy=80%). At 213-214 rows, we read that the “7 genes may bore no or low prognostic values for the response status of IFN-β treatment”. But in the Results section we do not find the performance of a machine learning model trained by using only these 7 genes so how can we comment about the prognostic value of these subset of 7 genes? This sentence in the discussion section is not supported by the results of the statistical analysis.

• Finally and more important, the work finds a significant difference in correlation between IL2RB and IL2RA and CXCR3 and CSF2. The authors conclude at 222-230 rows that “to determine which patients might respond to IFN-β treatment, instead investigating the expression values of these genes, the correlation directions between IL2RA and IL2RB, and CXCR3 and CSF2 should be focused. If the correlation directions are negative for the two gene pairs (meaning one is overexpressed / under-expressed, then the other tends to be overexpressed / under-expressed as well. For a single sample, the overexpression or under-expression may be determined by comparing with mean expression values of a reference population), the patient tends to respond to IFN-β treatment fairly well. For a patient without such negative correlation patterns among these two gene pairs, maybe an alternative regimen should be pursued.”

The authors suggest to use for a single patient this value (“ the correlation direction”) to predict the status of responder. This value is the contribution of the patient to the calculus of the correlation that can be computed as the average value of these contributions on the data set. The prediction of the response of the patient based on this value is in general an improperly posed question. In details, we can have significant difference of correlation also in the case of small correlation in a class and large value of correlation in the other class. The correlation is small when the values on single patients are small or large but with a mean value near zero. In the latter case, in the class with small correlation all patients have different values that will be also different in sign: how can you use the value on a patient to predict the response to the treatment?
In the case of large negative correlation between two genes in a class and large positive correlation in the other class, the question could be well posed and we can suppose that for a patient the direction of correlation (or better of its contribution to correlation) can help to predict the response of the patients but in this work the authors did not perform a statistical analysis to test the prognostic value of this variable. The result section refer to a machine learning model based on the value of the 41 genes and not on this “correlation direction” between pairs of genes. The result section is not useful to support this idea described in the discussion section.

Additional comments

I have a question about SNPs in IL2RA (254 row). Can your study suggest that SNPs related to MS but present only in one class (responder or not responder) can alterate the regulation of the expression of a gene in the DCE and produce the difference in the correlation with the other gene in the DCE?

---

## Round 0.3 · accepted · Accept

The last concerns raised by the reviewers have been addressed, thus I consider the paper ready for publication.